# Prognostic Impact of LAG-3 mRNA Expression in Early Breast Cancer

**DOI:** 10.3390/biomedicines10102656

**Published:** 2022-10-21

**Authors:** Anne-Sophie Heimes, Katrin Almstedt, Slavomir Krajnak, Anne Runkel, Annika Droste, Roxana Schwab, Kathrin Stewen, Antje Lebrecht, Marco J. Battista, Walburgis Brenner, Annette Hasenburg, Mathias Gehrmann, Jan G. Hengstler, Marcus Schmidt

**Affiliations:** 1Department of Gynecology and Obstetrics, University Medical Center Mainz, 55131 Mainz, Germany; 2Bayer AG, 42113 Wuppertal, Germany; 3Leibniz-Research Centre for Working Environment and Human Factors at the TU Dortmund (IfADo), 44139 Dortmund, Germany

**Keywords:** LAG-3, immune checkpoints, immunotherapy, breast cancer

## Abstract

**Background:** Monoclonal antibodies against PD-1 or PD-L1 have been established in clinical practice for the treatment of both early and advanced/metastatic triple-negative breast cancer. Beyond the established immune checkpoints (ICPs) (PD-1 and CTLA-4), additional ICPs, such as lymphocyte activation gene-3 (LAG-3), are subject of current research. In the present retrospective gene-expression analysis, we evaluated the prognostic significance of LAG-3 in 461 patients with early breast cancer. In addition, we examined whether there was a correlation between the different ICP and CD8 expressions. **Methods:** Using microarray-based gene-expression analysis, we examined the prognostic significance of LAG-3 mRNA expression for metastasis-free survival (MFS) in the whole cohort of 461 breast cancer patients and among different molecular subtypes. Correlations were analyzed using Spearman’s rho correlation coefficient. **Results:** In the whole cohort, LAG-3 expression had no significant impact on MFS (*p* = 0.712, log-rank). In the subgroup analyses, there was a trend that a higher LAG-3 expression was associated with a favorable outcome in the luminal B (*p* = 0.217), basal-like (*p* = 0.370) and HER2 (*p* = 0.089) subtypes, although significance was not reached. In contrast, in a multivariate Cox regression analysis, adjusted for age, tumor size, axillary nodal status, histological grade of differentiation and proliferation marker Ki-67, LAG-3 showed a significant influence on MFS (HR 0.574; 95% CI 0.369–0.894; *p* = 0.014). High LAG-3 significantly correlated with CD8 (ρ = 0.571; *p* < 0.001). **Conclusions:** LAG-3 expression had an independent impact on MFS. In addition to PD-1 and PD-L1, further immune checkpoints, such as LAG-3, could serve as therapeutic targets in breast cancer.

## 1. Introduction

Immunotherapies, such as immune checkpoint inhibitors (ICPi) or antibody drug conjugates (ADCs) [1], are of increasing importance in the development of targeted therapy strategies for various solid tumors, including breast cancer [2,3]. Breast cancer can be classified into several molecular subtypes (luminal A, luminal B, HER2-positive and triple-negative), which differ in their prognosis, as well as into different systemic therapy options (e.g., chemotherapy, endocrine therapy, anti-HER2 therapy) [4]. Triple-negative breast cancer (TNBC), which is defined by a lack of expression of estrogen, progesterone and HER2 (human epidermal growth factor receptor 2) receptors, accounts for 10–15% of all breast cancers and is characterized by a more aggressive tumor growth, a poorer grade of differentiation and a higher proliferation index (Ki-67), as well as a correspondingly poorer prognosis [5]. Due to a higher mutational load and a higher number of tumor-associated, immunogenic neoantigens, immunotherapeutic approaches play an important role in the development of targeted therapy strategies in triple-negative breast cancer. In addition to ADCs, T-cell-based approaches, in particular ICPi, are used in the treatment of TNBC [6]. ICPi block the interaction of specific cell-surface proteins of activated T-cells, which serve as negative regulators of T-cell-based immune responses, thereby enhancing the anti-tumor immune response. Currently, the most clinically relevant immune checkpoints (ICPs) include CTLA-4 (cytotoxic T-lymphocyte-associated protein 4) and, in breast cancer, the PD-1 (programmed cell-death protein 1)/PD-L1 (programmed death-ligand 1) axis. The rationale for the use of ICPi is to release a brake of the immune system, thereby increasing antitumor activity. Based on the convincing data of numerous phase III studies, immune checkpoint inhibitors, such as atezolizumab or pembrolizumab, have been used in clinical practice in early TNBC in combination with neoadjuvant cytotoxic therapy (NACT) [7,8], and in metastatic TNBC in combination with taxane-based chemotherapy [9,10,11]. Nevertheless, the clinical efficacy of ICPi (such as atezolizumab or pembrolizumab) in monotherapy has been modest in phase I trials of advanced and extensively pretreated TNBC [12]. Cases of non-response and primary or secondary resistance when using ICPi in monotherapy have also been observed in other tumor entities, such as high-grade serous ovarian cancer [13]. A possible explanation for the development of resistance could be the compensatory upregulation of other immune checkpoints once a single immune checkpoint is blocked by a monoclonal antibody. For example, Huang and colleagues demonstrated in an ovarian cancer mouse model that the inhibition of PD-1 by a monoclonal antibody led to the compensatory upregulation of other immune checkpoints, such as LAG-3 or CTLA-4 [14]. Another work by Huang et al. demonstrated in a preclinical murine ovarian cancer model that a dual blockade of LAG-3 and PD-1 had a synergistic effect in terms of anti-tumor immune response [15]. These data suggest the potential benefit of the combinatorial use of ICPi blocking multiple immune checkpoints. For other entities, such as melanoma or non-small-cell lung cancer, the clinical efficacy of combining different immune checkpoint inhibitors, particularly the combination of ipilimumab, a monoclonal antibody against CTLA-4 and nivolumab which blocks PD-1, has been demonstrated in phase III studies [16,17]. In addition, based on the promising data from a phase III study, the combination of relatlimab, an antibody blocking LAG-3, and nivolumab as a fixed-dose combination was recently approved by the FDA for the treatment of advanced melanoma as first-line therapy [18]. Therefore, the important role of “next generation” immune checkpoints, such as LAG-3, is obvious. LAG-3 is expressed on activated tumor-infiltrating lymphocytes (TILs) (for example CD4 and CD8 positive T-cells [19], T-regs [20,21] and B-lymphocytes), and it regulates T-cell function as an inhibitory ICP. In the present study, to better understand the importance of LAG-3 in breast cancer and its interaction and correlation with other immune checkpoints, such as CTLA-4 and PD-1, we evaluated the prognostic significance of LAG-3 by gene-expression analysis in a cohort of 461 breast cancer patients. In addition, we analyzed the prognostic impact of other immune checkpoints (CTLA-4, PD-1) and CD8 expression as a marker for activated cytotoxic T-cells. Furthermore, we examined whether there was a correlation between the different immune checkpoints and CD8 expression.

## 2. Materials and Methods

### 2.1. Patient’s Characteristics

The study cohort included 461 patients with early breast cancer who underwent surgery at the Department of Gynecology and Obstetrics at the University Medical Center Mainz between 1986 and 2000 and from whom sufficient tumor tissue (fresh frozen) was available for successful Affymetrix microarray analysis. The whole cohort consisted of three subgroups with different systemic treatments:

(i) “N0 cohort”: 200 node-negative patients with early breast cancer who received no further adjuvant therapy after surgery and radiation.

(ii) “Tamoxifen cohort”: 165 patients treated with tamoxifen as a single adjuvant therapy.

(iii) “Chemotherapy cohort”: 96 patients treated with either cyclophosphamide, methotrexate, fluorouracil (CMF; n = 34) or epirubicin and cyclophosphamide (EC; n = 62) in the adjuvant setting. The abovementioned chemotherapy regimens were applied as adjuvant therapy after the completion of surgical therapy, and thus had no effect on the analyzed mRNA levels.

The patient’s detailed characteristics are summarized in Table 1. Established prognostic factors (histologic grade of differentiation, tumor size, nodal status, age at initial diagnosis, ER, PR, HER2 and Ki-67) were obtained from pathology reports and the breast cancer database of our department. The median age of patients at initial diagnosis was 62 years. The median follow-up time was 12.75 years; 133 patients (28.9%) developed distant metastases.

This study was approved by the Ethics Committee of Rhineland-Palatinate, Germany [no. 837.139.05 (4797)]. Written informed consent was obtained from all patients, and all clinical investigations were conducted ethically in accordance with ethical and legal standards and in consideration of the Declarations of Helsinki.

### 2.2. mRNA Isolation and Gene-Expression Analysis

The mRNA isolation and gene-expression analysis were performed as previously described [22]. Briefly, tumor samples were frozen and then placed in storage at −80 °C. About 50 mg of the frozen breast tumor tissue was fragmented in liquid nitrogen. An RLT buffer was applied, and the resulting homogenate was centrifuged through a QIAshredder column (Qiagen). Total RNA was further isolated from the eluate using the RNeasy kit (Qiagen) according to the manufacturer’s instructions. RNA yield was determined by UV absorbance, and RNA quality was evaluated by rRNA-ban-integrity analysis using an Agilent 2100 Bioanalyzer RNA 6000 LabChip Kit (Agilent Technologies, Santa Clara, CA 95051, USA), as previously described [22].

For mRNA expression analysis, fresh frozen tumor tissue from a total of 461 breast cancer samples from the Department of Gynecology and Obstetrics at the University Medical Center Mainz was used to generate HG-U133A arrays (Affymetrix, Santa Clara, CA, USA) and to measure the relative transcript frequencies of the genes listed below in the tumor tissue. A total of 5 µg of total RNA, labeled cRNA, was prepared using the Roche Microarray cDNA Synthesis, Microarray RNA Target Synthesis (T7), and Microarray Target Purification Kits (Roche Applied Science, Mannheim, Germany) following the manufacturer’s instructions. Raw expression data (CEL files) were normalized by multiarray analysis (fRMA). Most samples were already deposited (2008 and 2011) with the National Center for Biotechnology Information (NCBI) Gene Expression Omnibus (GEO) under accession numbers GSE11121 and GSE26971. In addition, the complete dataset of the 461 samples used with an updated follow-up was previously deposited at the NCBI in the GEO database under accession number GSE158309 [22].

For gene-expression analysis, the following single genes were considered with the corresponding probe sets:LAG-3: 206486_atCTLA-4: 221331_x_atPD-1: 207634_atCD8: 205758_at

In addition, an immune checkpoint signature (CPS) was averaged from the gene-expression values of the single genes LAG-3, CTLA-4 and PD-1. For dichotomization, scores above the median of the CPS signature were defined as high expression, whereas scores below the median were defined as low expression.

### 2.3. Molecular Subtypes

Intrinsic subtypes were assessed according to Haibe-Kains et al. [23], who established a three-gene model, including the estrogen receptor gene (ESR1), HER2 and Aurora kinase A (AURKA). Briefly, the ER and HER2 status was determined from the bimodally distributed mRNA levels of the corresponding genes (probe sets: ESR1 205225_at and ERBB2 216836_s_at) based on fRMA-normalized expression values. The cut-off value for ESR1 was determined by model-based clustering. The cut-off value for ERBB2 was determined using the upper-quartile plus the interquartile range of the mRNA level. For AURKA, the median mRNA expression of the corresponding probe set (208079_s_at) was used as the cut-off value, as described previously [22]. This procedure resulted in the following molecular subtypes:−ESR1-positive, HER2-negative, low proliferation (AURKA low) luminal-A-like.−ESR1-positive, HER2-negative, high proliferation (AURKA high) luminal-B-like.−HER2-positive.−ESR1 negative HER2 negative basal-like.

Table 1 shows the absolute and relative frequencies of molecular subtypes determined by gene-expression data within the patient cohort studied.

To validate our gene-expression analyses on a larger, independent cohort, we used publicly available gene-expression data of LAG-3 with associated tumor characteristics, as well as follow-up data [24], in an unselected cohort of breast cancer patient, and in the subgroup of triple-negative breast cancer samples. 

### 2.4. Statistical Analysis

Statistical analyses were performed using the SPSS statistical software program, version 23.0 (SPSS Inc, Chicago, IL, USA) and Stata, Version 17 (StataCorp LLC Lakeway Drive, TX 77845-4512. USA). The prognostic significance of LAG-3, PD-1, CTLA-4 and CD8 expression for MFS was examined by Kaplan–Meier survival analysis (≤median vs. >median), as well as univariate and multivariate Cox regression analysis. The multivariate Cox regression analysis was adjusted for pT stage (T1 vs. T2, T3, 4), histological grade (GI + GII vs. GIII), Ki-67 (≤20% vs. >20%) and lymph node state (negative vs. positive). The significance of the Kaplan–Meier survival analysis was assessed by the *p*-value of the log-rank test. Correlations were analyzed using Spearman’s rho correlation coefficient.

## 3. Results

### 3.1. Prognostic Impact of LAG-3, CTLA-4, PD-1 and CD8 mRNA Expression

Kaplan–Meier analysis showed that, within the whole cohort, consisting of the N0, tamoxifen and chemotherapy subcohorts, LAG-3 expression had no significant impact on MFS (*p* = 0.712, log-rank) (Figure 1a). Within the subgroup analyses, there was a trend that a higher LAG-3 expression was associated with a longer MFS in the luminal B (*p* = 0.217, log-rank), basal-like (*p* = 0.370 log-rank) and HER2 (*p* = 0.089 log-rank) subtypes, although significance was not reached (Figure 1b). 

In contrast, in the multivariate Cox regression analysis adjusted for age, tumor size, axillary nodal status, histological grade of differentiation and the proliferation marker Ki-67, LAG-3 showed a significant influence on MFS (HR 0.574; 95% CI 0.369–0.894; *p* = 0.014) (Table 2), and a higher LAG-3 expression was associated with a longer MFS. In addition to LAG-3 expression, tumor grading (HR 2.583; 95% CI 1.591–4.192; *p* < 0.001) and tumor size (HR 1.626; 95% CI 1.020–2.591; *p* = 0.041) were also identified as independent prognostic factors in the multivariate Cox regression analysis (Table 2, Appendix A). Additionally, univariate and multivariate Cox regression analyses, adjusted for age at diagnosis, tumor size, nodal status, histological grade of differentiation and the proliferation marker Ki-67, were performed among the molecular subtypes showing a significant association between LAG-3 expression and MFS in the luminal-B-like subtype. (HR 0.504; 95% CI 0.258–0.985; *p* = 0.045) (Table 3).

Within the whole cohort, no significant effect of CTLA-4 expression regarding MFS could be demonstrated (*p* = 0.664 log-rank) (Appendix A). Among the molecular subtypes, there was a trend toward a better outcome, associated with a higher CTLA-4 expression in the basal-like subtype, although a significance level was not reached (*p* = 0.072 log-rank) (Appendix A). CTLA-4 expression failed to be proven as a significant prognostic factor in univariate and multivariate Cox regression analyses (Appendix A). Regarding PD-1 expression, there was no significant effect on MFS within the whole cohort (Appendix A) or within the molecular subtypes (Appendix A). With respect to CD8, Kaplan–Meier curves showed that CD8 expression had no significant influence on MFS, either in the whole cohort (Appendix A) or within subgroup analyses (Appendix A). In contrast, CD8 was a significant prognostic factor in the multivariate Cox regression analysis (HR 0.642; 95% CI 0.421–0.979; *p* = 0.040), in which a higher CD8 expression was associated with a better outcome (Table 4, Appendix A).

### 3.2. Validation of the Prognostic Impact of LAG-3 mRNA Expression in an Independent Cohort

To validate our gene-expression analyses in a larger, independent cohort, we used publicly available gene-expression data of LAG-3 with associated tumor characteristics, as well as follow-up data [24], in an unselected cohort of breast cancer patients (n = 4929), and in the triple-negative breast cancer subgroup. The prognostic significance of LAG-3 expression was first determined within the overall cohort regarding both RFS and OS. Among the whole cohort, LAG-3 expression failed to show a prognostic impact in terms of RFS (*p* = 0.17 log-rank, n = 4929), as well as OS (*p* = 0.54 Rog Rank; n = 1879) (Appendix A). In contrast, within the subgroup of triple-negative breast cancers, LAG-3 was shown to be a significant prognostic factor regarding RFS (*p* = 0.01, n = 335) and OS (*p* = 0.037, n = 132): a higher LAG-3 expression was associated with a better outcome (Appendix A).

### 3.3. Prognostic Impact of an Immune Checkpoint Associated Signature (CPS)

CPS, calculated by averaging the gene-expression values of LAG-3, CTLA-4 and PD-1, showed no significant effect on MFS in the whole cohort (*p* = 0.242 log-rank) (Figure 2a). However, subgroup analysis showed a significant effect in the basal-like subtype, with a higher expression of CPS associated with a longer MFS (*p* = 0.050 log-rank) (Figure 2b). Multivariate Cox regression analysis among the molecular subtypes identified CPS as an independent prognostic factor in the basal-like subtype (HR 0.195; 95% CI 0.043–0.88; *p* = 0.034) (Table 5).

### 3.4. Correlation Effects

Spearman’s rho correlation coefficient showed a strong, significant correlation between LAG-3 and CD8: a higher CD8 expression was associated with a higher LAG-3 expression (ρ = 0.571; *p* < 0.001). This significant correlation between CD8 and LAG-3 expression is illustrated by the scatter plot (Figure 3).

A less strong, although still significant, correlation was found between CTLA-4 expression and CD8 expression (ρ = 0.447; *p* < 0.001), which is shown in Figure 4.

In contrast, no significant correlation relationship was demonstrated between PD-1 expression and CD8. In addition, the correlation between the immune checkpoints was analyzed. In this context, a strong correlation between LAG-3 and CTLA-4 expression could be shown (ρ = 0.453; *p* < 0.001) (Figure 5).

A less strong, but still significant, correlation between LAG-3 and PD-1 was also demonstrated (ρ = 0.243; *p* < 0.001). In contrast, an inverse but significant correlation was found between CTLA-4 and PD-1 expression (ρ = −0.213; *p* < 0.001).

## 4. Discussion

In the present, retrospective, gene-expression study, we demonstrated that LAG-3, as a next generation ICP, has independent prognostic significance in terms of MFS in a cohort of 461 breast cancer patients with long-term follow up. Furthermore, we showed a significant correlation between LAG-3 expression and CD8 expression.

Immune checkpoint inhibitors are well established as targeted immunotherapies for the treatment of various solid tumors [25]. In breast cancer, the PD-1/PD-L1 axis is particularly important, so that atezolizumab, for example, a monoclonal antibody against PD-L1, in combination with nab-paclitaxel, is an approved therapeutic option in advanced triple-negative breast cancer [9]. In addition to atezolizumab, the monoclonal anti PD-1 antibody pembrolizumab in combination with taxane-containing chemotherapy is an approved treatment option for metastatic triple-negative breast cancer [11]. In early triple-negative breast cancer, recent phase III data have also demonstrated the efficacy of pembrolizumab in combination with neoadjuvant chemotherapy [7,26].

In contrast, ICPi monotherapy has been shown to be less effective in advanced TNBC. Additionally, primary or secondary resistance make it necessary to improve the clinical efficacy of ICPi. In this context, “next generation” immune checkpoints, such as LAG-3, play an increasing role. LAG-3 is frequently co-expressed with PD-1 on activated CD4- and CD8-positive tumor-infiltrating T-cells, and is known to be a negative regulator of T-cell function [27]. This is in line with our findings, showing a significant correlation between LAG-3 and PD-1 mRNA expression (ρ = 0.243; *p* < 0.001) in a cohort of 461 patients with early breast cancer. Furthermore, multivariate Cox regression identified LAG-3 as an independent prognostic factor in the whole cohort, showing that a higher LAG-3 expression was associated with a better MFS (HR 0.574; 95% CI 0.369–0.894; *p* = 0.014). One possible interpretation is that a higher mRNA expression of immune checkpoint regulators implies the presence of TILs associated with a better survival. Indeed, we demonstrated a strong correlation between CD8 expression and LAG-3 expression. These findings are in line with the results of a study published by Denkert et al., which analyzed immune-associated mRNA markers in tumor samples from patients of the Gepar-Sixto trial, demonstrating that immunosuppressive markers, such as PD-1, PD-L1, CTLA-4 and FoxP3, were positively associated with the presence of stromal TILs and correlated with treatment response [28,29]. Among the subgroup of basal-like breast cancers, there was a significant effect of a higher expression of CPS (calculated from LAG-3, CTLA-4 and PD-1) being associated with a longer MFS (*p* = 0.05, log-rank). This fits with the results of a previous gene-expression study, in which we were able to show that a CPS consisting of CTLA-4 and PD-1 had a significant impact in terms of MFS in a subset of triple-negative breast cancers [30]. In order to validate our hypothesis (that LAG-3 expression is associated with better outcome, especially in triple-negative breast cancer) we validated our results using an independent and larger cohort of triple-negative breast cancer samples with publicly available gene-expression data and existing clinical and follow-up data [24]. These validation results showed that a higher LAG-3 expression was associated with a better outcome in terms of both OS and RFS. These data are further supported by the work of Stovgaard et al., who examined LAG-3 expression using immunohistochemistry on 514 TNBC tumor sections, demonstrating that a higher LAG-3 expression at the protein level was associated with longer OS and RFS [31]. A study by Sarradin et al. investigated the effects of neoadjuvant chemotherapy on the composition of the so-called immune microenvironment in triple-negative breast cancers [32]. It was shown that LAG-3 expression evaluated by immunohistochemistry decreased significantly after neoadjuvant chemotherapy. LAG-3 expression, analyzed before and after completion of chemotherapy, had no significant effect on OS [32]. Regarding our retrospective gene-expression analysis, chemotherapy was applied as an adjuvant therapy after the completion of surgical therapy, and thus had no effect on the analyzed LAG-3 mRNA levels.

Furthermore, in the present work, we showed a positive and significant correlation between both LAG-3 and PD-1 expression, and LAG-3 and CTLA-4 expression. A possible approach to explain the development of resistance and, consequently, an inadequate response to ICPi therapy is the compensatory upregulation of alternative immune checkpoints once a single immune checkpoint is blocked by a monoclonal antibody. Indeed, Saleh and colleagues demonstrated in a triple-negative cell co-culture model that blocking PD-1 or PD-L1 resulted in the compensatory upregulation of LAG-3, TIM-3 and CTLA-4 in CD4-positive T-cells [33]. A preclinical mouse model by Woo et al. revealed a synergistic effect of the dual blockade of PD-1 and LAG-3, resulting in a good response rate [27]. This promising approach of the dual blockade of two immune checkpoints was investigated in a randomized three-arm phase II trial in advanced TNBC using the anti-LAG-3 antibody LAG525 (Leramilimab), either in combination with the anti-PD-1 antibody spartalizumab, with carboplatin or as triplet therapy (NCT03499899). Preliminary results indicated that the most sustained response rates were achieved in the study arm with triplet therapy (ORR 32.4%) at the price of increased side effects [34].

Further studies will show if a dual blockade of different immune checkpoints should also be recommended in breast cancer. 

A limitation of the present study is that it was conducted retrospectively and that it is unicentric. Another potential weakness of our study is that our analyses were limited to gene-expression data. Although it would be interesting and desirable to validate the expression of LAG-3 at the protein level in the same collective, e.g., by immunohistochemistry, possibly in combination with immunofluorescent co-localization studies between LAG-3 and CD8, there is not enough material left to perform such an analysis, as this is a very old collective. A strength of the present study is the consecutive inclusion of patients with a sufficient amount of fresh frozen tissue available for successful DNA microarray analysis with long-term follow-up and well-defined adjuvant treatment strategies.

## 5. Conclusions

In summary, the present study identified LAG-3 as an independent prognostic factor for improved MFS. In addition, we showed a significant correlation between CD8 and LAG-3 expression.

## Figures and Tables

**Figure 1 biomedicines-10-02656-f001:**
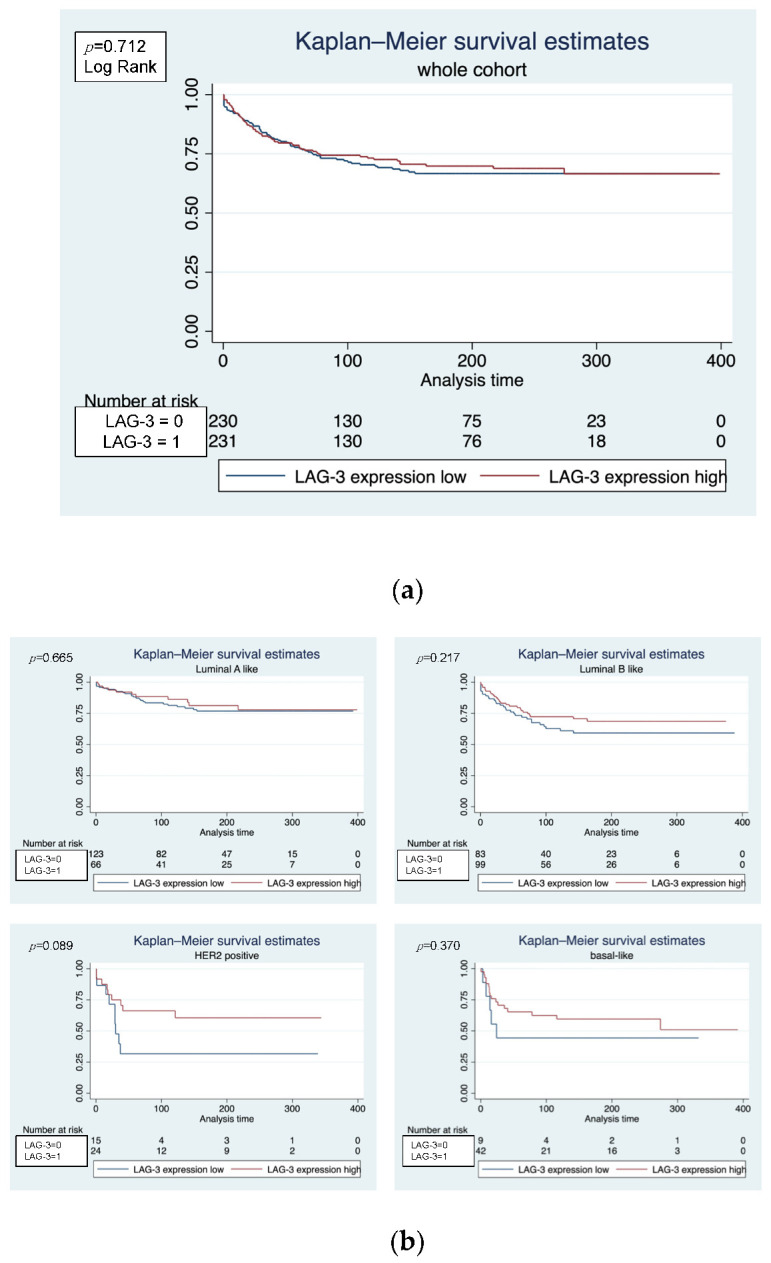
(**a**) Kaplan–Meier plot shows no significant effect of LAG-3 expression on MFS in the whole cohort. (**b**) Within the subgroup analyses, a trend is apparent that a higher LAG-3 expression was associated with a more favorable outcome (longer MFS) in the luminal B (*p* = 0.217, log-rank), basal-like (*p* = 0.370 log-rank) and HER2 (*p* = 0.089 log-rank) subtypes, although significance was not reached.

**Figure 2 biomedicines-10-02656-f002:**
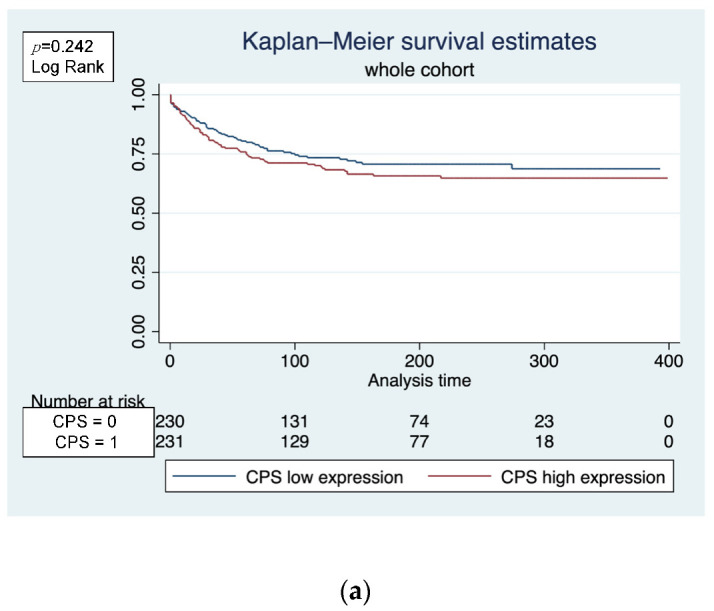
(**a**) CPS could not achieve significance for MFS in the whole cohort. (**b**) In contrast, there was a significant effect in the basal-like subtype (*p* = 0.050 log-rank).

**Figure 3 biomedicines-10-02656-f003:**
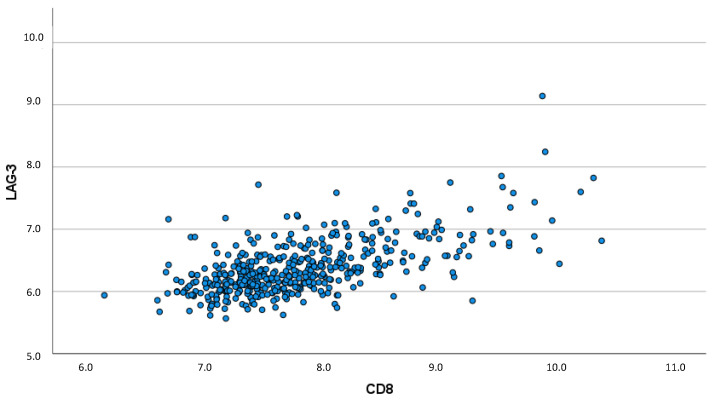
Scatter plot visualizing the correlation between CD8 and LAG-3 expression (ρ = 0.571; *p* < 0.001).

**Figure 4 biomedicines-10-02656-f004:**
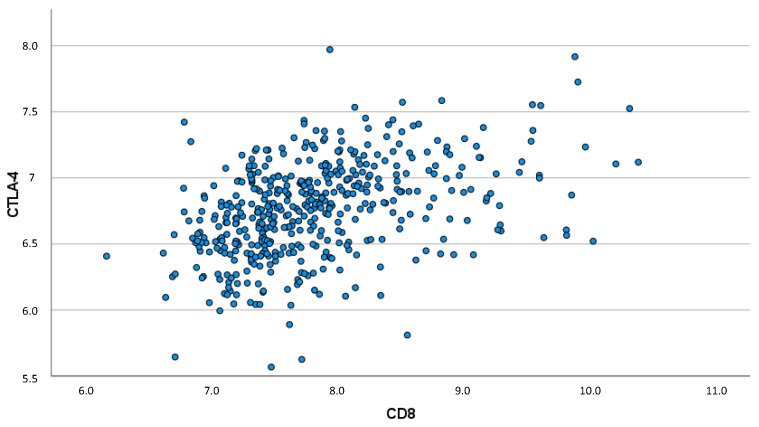
Scatter plot demonstrating the correlation between CD8 and CTLA-4 expression (ρ = 0.447; *p* < 0.001).

**Figure 5 biomedicines-10-02656-f005:**
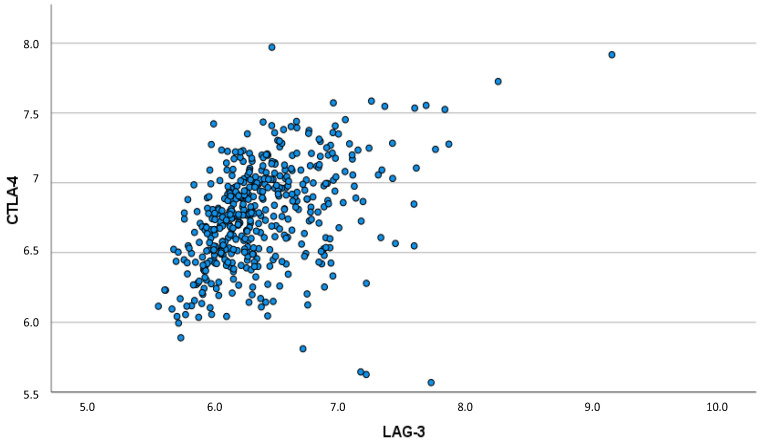
Scatter plot of correlation between CTLA-4 and LAG-3 expression (ρ = 0.453; *p* < 0.001).

**Table 1 biomedicines-10-02656-t001:** Patient’s characteristics (n = 461).

	Number of Patients (n = 461)	Percentage (%)
Age at diagnosis		
</=50 >50	104 357	22.6 77.4
Tumor size		
T1 T2 T3 T4 Missing value	188 214 19 39 1	40.8 46.4 4.1 8.5 0.2
Tumor grade		
GI GII GIII	63 287 111	13.7 62.3 24
Lymph node status		
N0 N1 N2 Nx	254 140 49 18	55.1 30.4 10.6 3.9
Tumor type		
Invasive ductal (NST) Invasive lobular Others	291 79 91	63.1 17.1 19.7
ER		
Positive Negative Missing value	381 79 1	82.6 17.1 0.2
PR		
Positive Negative Missing value	346 114 1	75.1 24.7 0.2
HER2		
Positive Negative Missing value	46 358 57	10 77.7 12.3
Ki-67		
>20% ≤20% Missing value	138 250 73	29.9 54.2 15.8
Molecular subtypes		
Lumina-A-like Luminal-B-like Basal-like HER2-positive	189 182 51 39	41 39.5 11.0 8.5
Distant metastasis		
Yes No	133 328	28.9 71.1
Treatment collective		
N0, untreated Tamoxifen Chemotherapy: CMF EC	200 165 96: 34 62	43.4 35.8 20.8: 7.4 13.4

**Table 2 biomedicines-10-02656-t002:** Multivariate Cox regression analysis of LAG-3 for MFS adjusted for age, tumor size, lymph node status, grade of differentiation and the proliferation marker Ki-67.

		HR	95% CI		*p*-Value
			Lower	Upper	
LAG3	High vs. low	0.574	0.369	0.894	0.014
Age	<50 vs. >50	1.176	0.681	2.031	0.561
Tumor size	T2–4 vs. T1	1.626	1.020	2.591	0.041
Lymph node status	N1,2,3 vs. N0	1.494	0.965	2.313	0.071
Grade	GIII vs. GI/II	2.583	1.591	4.192	<0.001
Ki-67	>20% vs. <20%	1.350	0.856	2.130	0.197

**Table 3 biomedicines-10-02656-t003:** Association between LAG-3 and MFS in molecular subtypes using univariate and multivariate Cox regression analysis adjusted for age at diagnosis, tumor size, nodal status, grading and Ki-67.

Subtype	Univariate Model HR (95% CI)	*p*-Value	Multivariate Model HR (95% CI)	*p*-Value
Luminal-A-like	0.855 (0.421–1.739)	0.666	0.614 (0.252–1.496)	0.283
Luminal-B-like	0.723 (0.430–1.216)	0.221	0.504 (0.258–0.985)	0.045
HER2-positive	0.455 (0.179–1.160)	0.099	0.276 (0.074–1.032)	0.056
Basal-like	0.638 (0.235–1.731)	0.377	0.459 (0.097–2.159)	0.324

**Table 4 biomedicines-10-02656-t004:** Multivariate Cox Regression analysis of CD8 for MFS adjusted for age, tumor size, lymph node status, grade of differentiation and the proliferation marker Ki-67.

		HR	95% CI		*p*-Value
			Lower	Upper	
CD8	High vs. Low	0.642	0.421	0.979	0.040
Age	<50 vs. ≥50	1.102	0.635	1.910	0.730
Tumor size	T2–4 vs. T1	1.529	0.959	2.436	0.074
Lymph node status	N1,2,3 vs. N0	1.486	0.958	2.395	0.077
Grade	GIII vs. GI/II	2.245	1.413	3.568	0.001
Ki-67	>20% vs. <20%	1.393	0.880	2.203	0.157

**Table 5 biomedicines-10-02656-t005:** Association between CPS and MFS in molecular subtypes using univariate and multivariate Cox regression analysis adjusted for age at diagnosis, tumor size, nodal status, grading and Ki-67.

Subtype	Univariate Model HR (95% CI)	*p*-Value	Multivariate Model HR (95% CI)	*p*-Value
Luminal-A-like	1.591 (0.824–3.071)	0.166	0.984 (0.426–2.273)	0.984
Luminal-B-like	0.948 (0.564–1.595)	0.841	0.536 (0.277–1.036)	0.064
HER2-positive	0.616 (0.237–1.604)	0.321	0.705 (0.219–2.274)	0.558
Basal-like	0.420 (0.171–1.034)	0.059	0.195 (0.043–0.881)	0.034

## Data Availability

Not applicable.

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
