# Peer review of "Prognostic Impact of LAG-3 mRNA Expression in Early Breast Cancer"

_biomedicines, 2022, doi:10.3390/biomedicines10102656_

Round 1

Reviewer 1 Report

Heimes et al have developed a work using the LAG-3 gene expression in a cohort with different subtypes of breast cancer to evaluate whether this gene could be used as a predictor for prognosis. The occurrence of metastases were evaluated, and the correlation between MFS and tumor subtypes and other tumor characteristics, such as size, patient age, Ki-67, among others, were analyzed. The authors mention that no significance was achieved in some of the cases. Also, a correlation between this gene and CD-8 was found, but not further explored. The paper ends with the conclusion that this gene should be used as a target for breast cancer.

Although the cohort used is a robust one, with many breast cancer patients, for some analyses authors inform that significance was not reached. Does it mean that the targets used in this study were not related to good or bad prognosis? Do these analyses need more samples to reach a good statistical result?

The text, although in a correct English, seems  too dense, which brings some difficulties of comprehension for readers. I suggest a revision with further explanation of the terms and the objectives throughout the text, to facilitate comprehension of the results.

Author Response

Point-by-point response to the reviewers’ comments

We appreciate the reviewers’ comments, and we have prepared a revised version of the

manuscript. Please find enclosed our responses to the reviewers’ comments and suggestions:

Reviewer 1

Heimes et al have developed a work using the LAG-3 gene expression in a cohort with different subtypes of breast cancer to evaluate whether this gene could be used as a predictor for prognosis. The occurrence of metastases were evaluated, and the correlation between MFS and tumor subtypes and other tumor characteristics, such as size, patient age, Ki-67, among others, were analyzed. The authors mention that no significance was achieved in some of the cases. Also, a correlation between this gene and CD-8 was found, but not further explored. The paper ends with the conclusion that this gene should be used as a target for breast cancer.

Comment 1: Although the cohort used is a robust one, with many breast cancer patients, for some analyses authors inform that significance was not reached. Does it mean that the targets used in this study were not related to good or bad prognosis? Do these analyses need more samples to reach a good statistical result?

Response 1: The reviewer raises important questions for the clarity and significance of the manuscript's findings. This retrospective gene expression analysis examines the prognostic significance of LAG-3 expression using a cohort including 461 breast cancer patients with long term follow-up. While in Kaplan Meier analysis and univariate Cox regression within the whole cohort no significant effect of LAG-3 expression with regard to MFS can be detected, in multivariate Cox regression analysis adjusted for tumor size, nodal status, grading and the proliferation index Ki67 LAG3 was identified as a significant prognostic factor. Immune checkpoints are of particular interest as therapeutic targets within the triple negative breast cancer subgroup. Our cohort contained 11% “basal like” breast cancer cases, representing a case number of n=51. Although a trend was found within this small group that higher LAG-3 expression was associated with better outcome, significance was not achieved. To verify that the lack of significance was indeed related to the too small case number, we validated our hypothesis on a larger, independent cohort (publicly available mRNA data of triple negative breast carcinoma cases), showing that in terms of both RFS and OS, higher LAG-3 expression is associated with a significantly longer RFS / OS. In contrast, this significant association could not be shown when considering all molecular subtypes and thus a cohort of 4929 breast carcinoma cases (mRNA data). We have added the above results for validation in the manuscript and shown the corresponding figures (Figure A6, A7) in the appendix.

Comment 2: The text, although in a correct English, seems too dense, which brings some difficulties of comprehension for readers. I suggest a revision with further explanation of the terms and the objectives throughout the text, to facilitate comprehension of the results.

Response 2: This is a useful recommendation. We have tried to present the results in a more understandable way. For a better comprehension and understanding of the technical terms used, we added a list of abbreviations.

Abbreviations:

ADC

Antibody drug conjuagte

CD8

cluster of differentiation 8

CPS

immuncheckpoint signature

CTLA-4

cytotoxic T-lymphocyte-associated protein 4

ER

Estrogen receptor

HER2

human epidermal growth factor receptor 2

HR

Hazard ratio

LAG-3

lymphocyte activation gene-3

ICP

immune checkpoints

ICPi

immune checkpoint inhibitors

MFS

Metastasis free survival

NACT

neoadjuvant cytotoxic therapy

OS

Overall survival

PD-1

Programmed cell death protein 1

PD-L1

Programmed death-ligand 1

PR

Progesteron receptor

RFS

Recurrence free survival

TILs

Tumor infiltrating lymphocytes

TNBC

Triple-negative breast cancer

Reviewer 2 Report

The manuscript by Heimes et al. provides interesting and important evidence on the prognostic value of the LAF-3 gene expression in breast cancer utilizing microarray analysis. The sample size included within this work is appropriate. However, other fundamental problems exist which makes this work not suitable for publication as a full original research in the current form.

- The sample was grouped based on different modes of adjuvant therapy (which is helpful for multivariant analysis), however, the sample analysis ignores modes of neoadjuvant therapy treatment (such as cytotoxic therapy) which can largely change the expression level of many of the investigated immune markers in this work.

- The work lacks confirmation of gene expression levels with further analysis of protein expression levels (potentially including co-localization immunofluorescent studies between LAG-3 and CD8)

- gene expression analysis of publicly available (independent) gene sets of other breast cancer cohorts is of value.

- several findings in this work demonstrated only a trend in support of the hypothesis but not robust statistical significance.

- the discussion is superficial and lacks rigorous discussion of the majority of literature on the prognostic importance of LAG-3 in breast cancer (or other cancer types) which necessary to explain some of the data that was not associated with statistical significance.

Author Response

Point-by-point response to the reviewers’ comments

We appreciate the reviewers’ comments, and we have prepared a revised version of the

manuscript. Please find enclosed our responses to the reviewers’ comments and suggestions:

Reviewer 2

The manuscript by Heimes et al. provides interesting and important evidence on the prognostic value of the LAG-3 gene expression in breast cancer utilizing microarray analysis. The sample size included within this work is appropriate. However, other fundamental problems exist which makes this work not suitable for publication as a full original research in the current form.

Comment 1: The sample was grouped based on different modes of adjuvant therapy (which is helpful for multivariant analysis), however, the sample analysis ignores modes of neoadjuvant therapy treatment (such as cytotoxic therapy) which can largely change the expression level of many of the investigated immune markers in this work.

Response 1: Basically, the aspect mentioned by the reviewer is very interesting and proven in the literature: Sarradin et al. demonstrated that the composition of the so-called immune microenvironment changes under neoadjuvant chemotherapy. Thus, it could be shown that the immunohistochemically evaluated LAG-3 expression decreased significantly after neoadjuvant chemotherapy.

In contrast, our collective included 165 patients who received adjuvant therapy with tamoxifen and an additional 96 patients who received adjuvant chemotherapy (CMF or EC). Both endocrine therapy and cytostatic therapy were applied after surgery, so it did not lead to an impact on the mRNA expression of the examined immune checkpoints or T-cell associated marker.

Comment 2: The work lacks confirmation of gene expression levels with further analysis of protein expression levels (potentially including co-localization immunofluorescent studies between LAG-3 and CD8)

Response 2: Thank you very much, this is a useful suggestion that would definitely enrich the present work. Since this is a very old collective, there is not enough material left to perform such analysis. Even though it would of course be interesting to validate LAG3 expression by, for example, immunohistochemistry at the protein level on the same collective.

Comment 3: gene expression analysis of publicly available (independent) gene sets of other breast cancer cohorts is of value.

Response 3: Thank you for the helpful suggestion. We validated our results on a larger, independent collective, identifying LAG-3 as a prognostic factor in terms of both RFS and OS within basal like tumors. We have added the following paragraph in the Results section:

To validate our gene expression analyses in a larger, independent cohort, we used publicly available gene expression data of LAG-3 with associated tumor characteristics and follow-up data [22] in an unselected cohort of breast cancer patients (n=4929) and in the triple-negative breast cancer subgroup. The prognostic significance of LAG-3 expression was first determined within the overall cohort regarding both RFS and OS. Among the whole cohort LAG-3 expression failed to show prognostic impact both in terms of RFS (p=0.17 Log Rank, n=4929) and OS (p=0.54 Log Rank; n=1879) (Figure A6a, b). In contrast, within the subgroup of triple-negative breast cancers, LAG-3 was shown to be a significant prognostic factor regarding RFS (p= 0.01, n=335) as well as OS (p=0.037, n=132): higher LAG-3 expression was associated with a better outcome (Figure A7a, b).

A)

B)

Figure A6: Kaplan Meier curves regarding LAG-3 mRNA expression in an unselected, independent cohort of breast cancer samples in terms of a) RFS and b) OS

a)

b)

Figure A7: Kaplan Meier curves regarding LAG-3 expression in the subgroup of triple negative breast cancer samples of an independent cohort of breast cancer in terms of a) RFS and b) OS

Comment 4: several findings in this work demonstrated only a trend in support of the hypothesis but not robust statistical significance.

Response 4: This observation is correct and the comment is therefore justified. As mentioned above, we added validation results from an independent cohort in our manuscript, which confirmed and thus strengthened our hypotheses that did not reach a significance level based on our gene expression data.

Comment 5: the discussion is superficial and lacks rigorous discussion of the majority of literature on the prognostic importance of LAG-3 in breast cancer (or other cancer types) which necessary to explain some of the data that was not associated with statistical significance.

Response 5: The comment is understandable. We therefore supplemented some works on the topic of prognostic significance of LAG -3 in triple negative breast cancers in the Discussion:

In order to validate our hypothesis, that LAG-3 expression is associated with better out-come, especially in triple negative breast cancer, we validated our results using an inde-pendent and larger cohort of triple negative breast cancer samples with publicly available gene expression data and existing clinical and follow-up data [24]. These validation results showed that higher LAG-3 expression was associated with a better outcome in terms of both OS and RFS. These data are further supported by a work of Stovgaard et al. who examined LAG-3 expression using immunohistochemistry on 514 TNBC tumor sections and demonstrated that higher LAG-3 expression at the protein level was associated with longer OS and RFS [31]. A study by Sarradin et al. investigated the effects of neoadjuvant chemotherapy on the composition of the so-called immune microenvironment in triple negative breast cancers [32]. It was shown that LAG-3 expression evaluated by immuno-histochemistry decreased significantly after neoadjuvant chemotherapy. LAG-3 expression analyzed before and after completion of chemotherapy had no significant effect on OS [32].

Reviewer 3 Report

This work describes study the correlation between LAG-3 and metastasis-free survival (MFS) in breast cancer, and the correlation between LAG-3 and CD8 expression.

This study is interesting and has great potential, but also has several flaws:

Minor revisions:

·         Ethical committee information and patient/participant consent information for patient’s samples should be stated on the Materials and Methods’ section.

·         The authors say: “Further studies will show, if dual blockade of different immune checkpoints should also be recommended in breast cancer.” However, these studies are already being conducted on preclinical and clinical levels. The introduction and discussion section should cite clinical trials that are targeting LAG-3 for breast cancer treatment. That would give strength to the discussion and conclusions. antiPD-1+antiLAG-3 combination have showed encouraging results in many clinical trials. In addition, the first combo (opdualag) has been recently approved by the FDA for clinical use. This should be presented in the introduction.

Major revisions:

·         My main concern with this manuscript is the lack of novelty, as most of the presented results are not significant, and the correlation between LAG-3/CD8, LAG-3/CTLA-4 and PD-1/LAG-3 has already been described for breast cancer and for many other tumours. I suggest the authors to re-write the paper putting the focus on the novel findings.

·         A limitation of the gene expression analysis of your samples is that only 4 single genes were considered for the analysis. It would be interesting if you could study more genes to find more novel correlations.

·         Being a retrospectively study, it can be challenging to perform more experimental validations on your data, as the authors discuss. However, it would give more strength to your results to validate them on public omics data, as it would serve as an additional validation. The Gene set enrichment analysis and the validation of the prognostic risk model could be complemented analysing the data with other public databases, which would serve as an additional validation and would give more strength to the conclusions. For example, TIMER2.0 – Cistrome could be used for systematically analysis of immune infiltrates across breast cancer and other cancer types. It would allow the authors to analyse the gene expression for the studied markers markers between tumour and normal, association between gene expression and clinical outcome; correlation between genes, association between immune infiltrates and gene expression; etc.

·         Do the authors have access to a QIAGEN Ingenuity Pathway Analysis (IPA) software license? If so, this tool would be very useful to functional analyse, integrate, and further understand data. It has a search capability for information on genes and allows interactive building of networks to represent biological systems. It provides detailed predicted biological activity of your dataset about upstream regulators, causal networks, canonical pathways, multi-levelled regulatory networks, mechanistic networks, downstream effects analysis, tox lists and tox functions, disease, and treatment conditions, among other things. If not, the analysis of the molecular and functional relevance of these genes could be enriched by using some pathway analysis free resources, such as Metascape, Reactome or STRING. Metascape is a tool designed to provide a comprehensive gene list annotation and analysis resource, that can be used to obtain network of enriched terms, top-level Gene Ontology biological processes and interaction networks among other tools. STRING however is a database of known and predicted protein-protein interactions. The interactions include direct (physical) and indirect (functional) associations and could provide interesting hypothetical information for these identified genes at a protein level.

·         The authors indicate: “In contrast, in multivariate Cox regression analysis adjusted for age, tumor size, ax-190 illary nodal status, histological grade of differentiation and the proliferation marker Ki-191 67, LAG-3 showed a significant influence on MFS (HR 0.574; 95% CI 0.369–0.894; p = 0.014) 192 (Table 2), a higher LAG-3 expression was associated with a longer MFS.”. Could you show these results as a Figure in addition of the table? It seems that in this sub-section only non-significant results are presented as figures.

Author Response

Point-by-point response to the reviewers’ comments

We appreciate the reviewers’ comments, and we have prepared a revised version of the

manuscript. Please find enclosed our responses to the reviewers’ comments and suggestions:

Reviewer 3

This work describes study the correlation between LAG-3 and metastasis-free survival (MFS) in breast cancer, and the correlation between LAG-3 and CD8 expression.

This study is interesting and has great potential, but also has several flaws:

Minor revisions:

Comment 1: Ethical committee information and patient/participant consent information for patient’s samples should be stated on the Materials and Methods’ section.

Response 1: thank you for the comment. We have extended / supplemented the already existing paragraph in the material and methods part. We added the following paragraph in the section “Material and Methods”:

The study was approved by the Ethics Committee of Rhineland-Palatinate, Germany [no. 837.139.05 (4797)]. Written informed consent was obtained from all patients, and all clinical investigations were conducted ethically in accordance with ethical and legal standards and in consideration of the Declarations of Helsinki.

Comment 2: The authors say: “Further studies will show, if dual blockade of different immune checkpoints should also be recommended in breast cancer.” However, these studies are already being conducted on preclinical and clinical levels. The introduction and discussion section should cite clinical trials that are targeting LAG-3 for breast cancer treatment. That would give strength to the discussion and conclusions. antiPD-1+antiLAG-3 combination have showed encouraging results in many clinical trials. In addition, the first combo (opdualag) has been recently approved by the FDA for clinical use. This should be presented in the introduction.

 Response 2: Thank you for the useful comment. We have supplemented the introduction with the following paragraph:

In addition, based on the promising data from a Phase III study, the combination of relat-limab, an antibody blocking LAG-3 and nivolumab as a fixed-dose combination was recently approved by FDA for the treatment of advanced melanoma as first-line therapy [18].

Major revisions:

Comment 3: My main concern with this manuscript is the lack of novelty, as most of the presented results are not significant, and the correlation between LAG-3/CD8, LAG-3/CTLA-4 and PD-1/LAG-3 has already been described for breast cancer and for many other tumours. I suggest the authors to re-write the paper putting the focus on the novel findings.

Response 3: The reviewer is absolutely correct in that the correlation between LAG-3/CD8, LAG-3/CTLA-4 and PD-1/LAG-3 has already been described in cancer. Nevertheless, our work adds to the existing literature in that we investigated the prognostic significance of LAG-3 mRNA expression in a collective with long follow-up and different molecular subtypes. Furthermore, we report a cohort in which almost half of the patients had not been treated in the adjuvant setting, which allows a clearer estimation of the prognostic significance. For this reason, we prefer to stay with the current and strongly prognosis-driven orientation of our manuscript.

Comment 4: A limitation of the gene expression analysis of your samples is that only 4 single genes were considered for the analysis. It would be interesting if you could study more genes to find more novel correlations.

Response 4: Thanks for this meaningful comment. The reviewer is correct that one of the limitations of our gene expression study is that only 4 genes are studied. In this gene expression analyses we focused on LAG-3 expression. We will evaluate other related genes in further gene expression studies.

Comment 5: Being a retrospectively study, it can be challenging to perform more experimental validations on your data, as the authors discuss. However, it would give more strength to your results to validate them on public omics data, as it would serve as an additional validation. The Gene set enrichment analysis and the validation of the prognostic risk model could be complemented analysing the data with other public databases, which would serve as an additional validation and would give more strength to the conclusions. For example, TIMER2.0 – Cistrome could be used for systematically analysis of immune infiltrates across breast cancer and other cancer types. It would allow the authors to analyse the gene expression for the studied markers markers between tumour and normal, association between gene expression and clinical outcome; correlation between genes, association between immune infiltrates and gene expression; etc.

Response 5: This is a really helpful and valuable comment. In the meantime, we validated the results of our gene expression analyses on a larger, independent cohort and were able to confirm that LAG-3 is a significant prognostic factor in terms of both RFS and OS in the subgroup of triple-negative breast cancers. We have added these validation results to the manuscript to better confirm our initial results:

To validate our gene expression analyses in a larger, independent cohort, we used publicly available gene expression data of LAG-3 with associated tumor characteristics and follow-up data [22] in an unselected cohort of breast cancer patients (n=4929) and in the triple-negative breast cancer subgroup. The prognostic significance of LAG-3 expression was first determined within the overall cohort regarding both RFS and OS. Among the whole cohort LAG-3 expression failed to show prognostic impact both in terms of RFS (p=0.17 Log Rank, n=4929) and OS (p=0.54 Rog Rank; n=1879) (Figure A6a, b). In contrast, within the subgroup of triple-negative breast cancers, LAG-3 was shown to be a significant prognostic factor regarding RFS (p= 0.01, n=335) as well as OS (p=0.037, n=132): higher LAG-3 expression was associated with a better outcome (Figure A7a, b).

Comment 6: Do the authors have access to a QIAGEN Ingenuity Pathway Analysis (IPA) software license? If so, this tool would be very useful to functional analyse, integrate, and further understand data. It has a search capability for information on genes and allows interactive building of networks to represent biological systems. It provides detailed predicted biological activity of your dataset about upstream regulators, causal networks, canonical pathways, multi-levelled regulatory networks, mechanistic networks, downstream effects analysis, tox lists and tox functions, disease, and treatment conditions, among other things. If not, the analysis of the molecular and functional relevance of these genes could be enriched by using some pathway analysis free resources, such as Metascape, Reactome or STRING. Metascape is a tool designed to provide a comprehensive gene list annotation and analysis resource, that can be used to obtain network of enriched terms, top-level Gene Ontology biological processes and interaction networks among other tools. STRING however is a database of known and predicted protein-protein interactions. The interactions include direct (physical) and indirect (functional) associations and could provide interesting hypothetical information for these identified genes at a protein level.

Response 6: thank you for this helpful suggestion. Our manuscript had a focus on the prognostic significance of LAG-3 expression within the overall cohort, as well as within the subgroup of basal-like breast caancers in particular. Our results were validated using an independent, larger cohort (see above). Furthermore, the question of correlations between LAG-3 and the T-cell marker CD8 on the one hand and between LAG-3 and other immune checkpoints (PD-1, CTLA-4) on the other hand was answered. For further assessment of the significance of LAG-3 within the immune microenvironment it may be helpful to investigate downstream effects and mechanistic networks using the proposed tools. This is without question very important, but it would clearly go beyond the scope of our present manuscript.

Comment 7: The authors indicate: “In contrast, in multivariate Cox regression analysis adjusted for age, tumor size, ax-190 illary nodal status, histological grade of differentiation and the proliferation marker Ki-191 67, LAG-3 showed a significant influence on MFS (HR 0.574; 95% CI 0.369–0.894; p = 0.014) 192 (Table 2), a higher LAG-3 expression was associated with a longer MFS.”. Could you show these results as a Figure in addition of the table? It seems that in this sub-section only non-significant results are presented as figures.

Response 7: Thank you for the helpful suggestion. We have presented the data resulting from multivariate Cox regression analyses as forest plots and added them to the manuscript accordingly.

Figure A1: Forest Plot showing LAG-3 as an independent prognostic factor in terms of MFS in multivariate Cox regression analysis adjusted for tumor size, lymph node status, tumor grade, age and the proliferation marker Ki67.

Figure A5: Forest Plot showing CD8 as an independent prognostic factor in terms of MFS in multivariate Cox regression analysis adjusted for tumor size, lymph node status, tumor grade, age and the proliferation marker Ki67.

Point-by-point response to the reviewers’ comments

We appreciate the reviewers’ comments, and we have prepared a revised version of the

manuscript. Please find enclosed our responses to the reviewers’ comments and suggestions:

Reviewer 3

This work describes study the correlation between LAG-3 and metastasis-free survival (MFS) in breast cancer, and the correlation between LAG-3 and CD8 expression.

This study is interesting and has great potential, but also has several flaws:

Minor revisions:

Comment 1: Ethical committee information and patient/participant consent information for patient’s samples should be stated on the Materials and Methods’ section.

Response 1: thank you for the comment. We have extended / supplemented the already existing paragraph in the material and methods part. We added the following paragraph in the section “Material and Methods”:

The study was approved by the Ethics Committee of Rhineland-Palatinate, Germany [no. 837.139.05 (4797)]. Written informed consent was obtained from all patients, and all clinical investigations were conducted ethically in accordance with ethical and legal standards and in consideration of the Declarations of Helsinki.

Comment 2: The authors say: “Further studies will show, if dual blockade of different immune checkpoints should also be recommended in breast cancer.” However, these studies are already being conducted on preclinical and clinical levels. The introduction and discussion section should cite clinical trials that are targeting LAG-3 for breast cancer treatment. That would give strength to the discussion and conclusions. antiPD-1+antiLAG-3 combination have showed encouraging results in many clinical trials. In addition, the first combo (opdualag) has been recently approved by the FDA for clinical use. This should be presented in the introduction.

 Response 2: Thank you for the useful comment. We have supplemented the introduction with the following paragraph:

In addition, based on the promising data from a Phase III study, the combination of relat-limab, an antibody blocking LAG-3 and nivolumab as a fixed-dose combination was recently approved by FDA for the treatment of advanced melanoma as first-line therapy [18].

Major revisions:

Comment 3: My main concern with this manuscript is the lack of novelty, as most of the presented results are not significant, and the correlation between LAG-3/CD8, LAG-3/CTLA-4 and PD-1/LAG-3 has already been described for breast cancer and for many other tumours. I suggest the authors to re-write the paper putting the focus on the novel findings.

Response 3: The reviewer is absolutely correct in that the correlation between LAG-3/CD8, LAG-3/CTLA-4 and PD-1/LAG-3 has already been described in cancer. Nevertheless, our work adds to the existing literature in that we investigated the prognostic significance of LAG-3 mRNA expression in a collective with long follow-up and different molecular subtypes. Furthermore, we report a cohort in which almost half of the patients had not been treated in the adjuvant setting, which allows a clearer estimation of the prognostic significance. For this reason, we prefer to stay with the current and strongly prognosis-driven orientation of our manuscript.

Comment 4: A limitation of the gene expression analysis of your samples is that only 4 single genes were considered for the analysis. It would be interesting if you could study more genes to find more novel correlations.

Response 4: Thanks for this meaningful comment. The reviewer is correct that one of the limitations of our gene expression study is that only 4 genes are studied. In this gene expression analyses we focused on LAG-3 expression. We will evaluate other related genes in further gene expression studies.

Comment 5: Being a retrospectively study, it can be challenging to perform more experimental validations on your data, as the authors discuss. However, it would give more strength to your results to validate them on public omics data, as it would serve as an additional validation. The Gene set enrichment analysis and the validation of the prognostic risk model could be complemented analysing the data with other public databases, which would serve as an additional validation and would give more strength to the conclusions. For example, TIMER2.0 – Cistrome could be used for systematically analysis of immune infiltrates across breast cancer and other cancer types. It would allow the authors to analyse the gene expression for the studied markers markers between tumour and normal, association between gene expression and clinical outcome; correlation between genes, association between immune infiltrates and gene expression; etc.

Response 5: This is a really helpful and valuable comment. In the meantime, we validated the results of our gene expression analyses on a larger, independent cohort and were able to confirm that LAG-3 is a significant prognostic factor in terms of both RFS and OS in the subgroup of triple-negative breast cancers. We have added these validation results to the manuscript to better confirm our initial results:

To validate our gene expression analyses in a larger, independent cohort, we used publicly available gene expression data of LAG-3 with associated tumor characteristics and follow-up data [22] in an unselected cohort of breast cancer patients (n=4929) and in the triple-negative breast cancer subgroup. The prognostic significance of LAG-3 expression was first determined within the overall cohort regarding both RFS and OS. Among the whole cohort LAG-3 expression failed to show prognostic impact both in terms of RFS (p=0.17 Log Rank, n=4929) and OS (p=0.54 Rog Rank; n=1879) (Figure A6a, b). In contrast, within the subgroup of triple-negative breast cancers, LAG-3 was shown to be a significant prognostic factor regarding RFS (p= 0.01, n=335) as well as OS (p=0.037, n=132): higher LAG-3 expression was associated with a better outcome (Figure A7a, b).

Comment 6: Do the authors have access to a QIAGEN Ingenuity Pathway Analysis (IPA) software license? If so, this tool would be very useful to functional analyse, integrate, and further understand data. It has a search capability for information on genes and allows interactive building of networks to represent biological systems. It provides detailed predicted biological activity of your dataset about upstream regulators, causal networks, canonical pathways, multi-levelled regulatory networks, mechanistic networks, downstream effects analysis, tox lists and tox functions, disease, and treatment conditions, among other things. If not, the analysis of the molecular and functional relevance of these genes could be enriched by using some pathway analysis free resources, such as Metascape, Reactome or STRING. Metascape is a tool designed to provide a comprehensive gene list annotation and analysis resource, that can be used to obtain network of enriched terms, top-level Gene Ontology biological processes and interaction networks among other tools. STRING however is a database of known and predicted protein-protein interactions. The interactions include direct (physical) and indirect (functional) associations and could provide interesting hypothetical information for these identified genes at a protein level.

Response 6: thank you for this helpful suggestion. Our manuscript had a focus on the prognostic significance of LAG-3 expression within the overall cohort, as well as within the subgroup of basal-like breast caancers in particular. Our results were validated using an independent, larger cohort (see above). Furthermore, the question of correlations between LAG-3 and the T-cell marker CD8 on the one hand and between LAG-3 and other immune checkpoints (PD-1, CTLA-4) on the other hand was answered. For further assessment of the significance of LAG-3 within the immune microenvironment it may be helpful to investigate downstream effects and mechanistic networks using the proposed tools. This is without question very important, but it would clearly go beyond the scope of our present manuscript.

Comment 7: The authors indicate: “In contrast, in multivariate Cox regression analysis adjusted for age, tumor size, ax-190 illary nodal status, histological grade of differentiation and the proliferation marker Ki-191 67, LAG-3 showed a significant influence on MFS (HR 0.574; 95% CI 0.369–0.894; p = 0.014) 192 (Table 2), a higher LAG-3 expression was associated with a longer MFS.”. Could you show these results as a Figure in addition of the table? It seems that in this sub-section only non-significant results are presented as figures.

Response 7: Thank you for the helpful suggestion. We have presented the data resulting from multivariate Cox regression analyses as forest plots and added them to the manuscript accordingly.

Figure A1: Forest Plot showing LAG-3 as an independent prognostic factor in terms of MFS in multivariate Cox regression analysis adjusted for tumor size, lymph node status, tumor grade, age and the proliferation marker Ki67.

Figure A5: Forest Plot showing CD8 as an independent prognostic factor in terms of MFS in multivariate Cox regression analysis adjusted for tumor size, lymph node status, tumor grade, age and the proliferation marker Ki67.

Point-by-point response to the reviewers’ comments

We appreciate the reviewers’ comments, and we have prepared a revised version of the

manuscript. Please find enclosed our responses to the reviewers’ comments and suggestions:

Reviewer 3

This work describes study the correlation between LAG-3 and metastasis-free survival (MFS) in breast cancer, and the correlation between LAG-3 and CD8 expression.

This study is interesting and has great potential, but also has several flaws:

Minor revisions:

Comment 1: Ethical committee information and patient/participant consent information for patient’s samples should be stated on the Materials and Methods’ section.

Response 1: thank you for the comment. We have extended / supplemented the already existing paragraph in the material and methods part. We added the following paragraph in the section “Material and Methods”:

The study was approved by the Ethics Committee of Rhineland-Palatinate, Germany [no. 837.139.05 (4797)]. Written informed consent was obtained from all patients, and all clinical investigations were conducted ethically in accordance with ethical and legal standards and in consideration of the Declarations of Helsinki.

Comment 2: The authors say: “Further studies will show, if dual blockade of different immune checkpoints should also be recommended in breast cancer.” However, these studies are already being conducted on preclinical and clinical levels. The introduction and discussion section should cite clinical trials that are targeting LAG-3 for breast cancer treatment. That would give strength to the discussion and conclusions. antiPD-1+antiLAG-3 combination have showed encouraging results in many clinical trials. In addition, the first combo (opdualag) has been recently approved by the FDA for clinical use. This should be presented in the introduction.

 Response 2: Thank you for the useful comment. We have supplemented the introduction with the following paragraph:

In addition, based on the promising data from a Phase III study, the combination of relat-limab, an antibody blocking LAG-3 and nivolumab as a fixed-dose combination was recently approved by FDA for the treatment of advanced melanoma as first-line therapy [18].

Major revisions:

Comment 3: My main concern with this manuscript is the lack of novelty, as most of the presented results are not significant, and the correlation between LAG-3/CD8, LAG-3/CTLA-4 and PD-1/LAG-3 has already been described for breast cancer and for many other tumours. I suggest the authors to re-write the paper putting the focus on the novel findings.

Response 3: The reviewer is absolutely correct in that the correlation between LAG-3/CD8, LAG-3/CTLA-4 and PD-1/LAG-3 has already been described in cancer. Nevertheless, our work adds to the existing literature in that we investigated the prognostic significance of LAG-3 mRNA expression in a collective with long follow-up and different molecular subtypes. Furthermore, we report a cohort in which almost half of the patients had not been treated in the adjuvant setting, which allows a clearer estimation of the prognostic significance. For this reason, we prefer to stay with the current and strongly prognosis-driven orientation of our manuscript.

Comment 4: A limitation of the gene expression analysis of your samples is that only 4 single genes were considered for the analysis. It would be interesting if you could study more genes to find more novel correlations.

Response 4: Thanks for this meaningful comment. The reviewer is correct that one of the limitations of our gene expression study is that only 4 genes are studied. In this gene expression analyses we focused on LAG-3 expression. We will evaluate other related genes in further gene expression studies.

Comment 5: Being a retrospectively study, it can be challenging to perform more experimental validations on your data, as the authors discuss. However, it would give more strength to your results to validate them on public omics data, as it would serve as an additional validation. The Gene set enrichment analysis and the validation of the prognostic risk model could be complemented analysing the data with other public databases, which would serve as an additional validation and would give more strength to the conclusions. For example, TIMER2.0 – Cistrome could be used for systematically analysis of immune infiltrates across breast cancer and other cancer types. It would allow the authors to analyse the gene expression for the studied markers markers between tumour and normal, association between gene expression and clinical outcome; correlation between genes, association between immune infiltrates and gene expression; etc.

Response 5: This is a really helpful and valuable comment. In the meantime, we validated the results of our gene expression analyses on a larger, independent cohort and were able to confirm that LAG-3 is a significant prognostic factor in terms of both RFS and OS in the subgroup of triple-negative breast cancers. We have added these validation results to the manuscript to better confirm our initial results:

To validate our gene expression analyses in a larger, independent cohort, we used publicly available gene expression data of LAG-3 with associated tumor characteristics and follow-up data [22] in an unselected cohort of breast cancer patients (n=4929) and in the triple-negative breast cancer subgroup. The prognostic significance of LAG-3 expression was first determined within the overall cohort regarding both RFS and OS. Among the whole cohort LAG-3 expression failed to show prognostic impact both in terms of RFS (p=0.17 Log Rank, n=4929) and OS (p=0.54 Rog Rank; n=1879) (Figure A6a, b). In contrast, within the subgroup of triple-negative breast cancers, LAG-3 was shown to be a significant prognostic factor regarding RFS (p= 0.01, n=335) as well as OS (p=0.037, n=132): higher LAG-3 expression was associated with a better outcome (Figure A7a, b).

Comment 6: Do the authors have access to a QIAGEN Ingenuity Pathway Analysis (IPA) software license? If so, this tool would be very useful to functional analyse, integrate, and further understand data. It has a search capability for information on genes and allows interactive building of networks to represent biological systems. It provides detailed predicted biological activity of your dataset about upstream regulators, causal networks, canonical pathways, multi-levelled regulatory networks, mechanistic networks, downstream effects analysis, tox lists and tox functions, disease, and treatment conditions, among other things. If not, the analysis of the molecular and functional relevance of these genes could be enriched by using some pathway analysis free resources, such as Metascape, Reactome or STRING. Metascape is a tool designed to provide a comprehensive gene list annotation and analysis resource, that can be used to obtain network of enriched terms, top-level Gene Ontology biological processes and interaction networks among other tools. STRING however is a database of known and predicted protein-protein interactions. The interactions include direct (physical) and indirect (functional) associations and could provide interesting hypothetical information for these identified genes at a protein level.

Response 6: thank you for this helpful suggestion. Our manuscript had a focus on the prognostic significance of LAG-3 expression within the overall cohort, as well as within the subgroup of basal-like breast caancers in particular. Our results were validated using an independent, larger cohort (see above). Furthermore, the question of correlations between LAG-3 and the T-cell marker CD8 on the one hand and between LAG-3 and other immune checkpoints (PD-1, CTLA-4) on the other hand was answered. For further assessment of the significance of LAG-3 within the immune microenvironment it may be helpful to investigate downstream effects and mechanistic networks using the proposed tools. This is without question very important, but it would clearly go beyond the scope of our present manuscript.

Comment 7: The authors indicate: “In contrast, in multivariate Cox regression analysis adjusted for age, tumor size, ax-190 illary nodal status, histological grade of differentiation and the proliferation marker Ki-191 67, LAG-3 showed a significant influence on MFS (HR 0.574; 95% CI 0.369–0.894; p = 0.014) 192 (Table 2), a higher LAG-3 expression was associated with a longer MFS.”. Could you show these results as a Figure in addition of the table? It seems that in this sub-section only non-significant results are presented as figures.

Response 7: Thank you for the helpful suggestion. We have presented the data resulting from multivariate Cox regression analyses as forest plots and added them to the manuscript accordingly.

Figure A1: Forest Plot showing LAG-3 as an independent prognostic factor in terms of MFS in multivariate Cox regression analysis adjusted for tumor size, lymph node status, tumor grade, age and the proliferation marker Ki67.

Figure A5: Forest Plot showing CD8 as an independent prognostic factor in terms of MFS in multivariate Cox regression analysis adjusted for tumor size, lymph node status, tumor grade, age and the proliferation marker Ki67.

Point-by-point response to the reviewers’ comments

We appreciate the reviewers’ comments, and we have prepared a revised version of the

manuscript. Please find enclosed our responses to the reviewers’ comments and suggestions:

Reviewer 3

This work describes study the correlation between LAG-3 and metastasis-free survival (MFS) in breast cancer, and the correlation between LAG-3 and CD8 expression.

This study is interesting and has great potential, but also has several flaws:

Minor revisions:

Comment 1: Ethical committee information and patient/participant consent information for patient’s samples should be stated on the Materials and Methods’ section.

Response 1: thank you for the comment. We have extended / supplemented the already existing paragraph in the material and methods part. We added the following paragraph in the section “Material and Methods”:

The study was approved by the Ethics Committee of Rhineland-Palatinate, Germany [no. 837.139.05 (4797)]. Written informed consent was obtained from all patients, and all clinical investigations were conducted ethically in accordance with ethical and legal standards and in consideration of the Declarations of Helsinki.

Comment 2: The authors say: “Further studies will show, if dual blockade of different immune checkpoints should also be recommended in breast cancer.” However, these studies are already being conducted on preclinical and clinical levels. The introduction and discussion section should cite clinical trials that are targeting LAG-3 for breast cancer treatment. That would give strength to the discussion and conclusions. antiPD-1+antiLAG-3 combination have showed encouraging results in many clinical trials. In addition, the first combo (opdualag) has been recently approved by the FDA for clinical use. This should be presented in the introduction.

 Response 2: Thank you for the useful comment. We have supplemented the introduction with the following paragraph:

In addition, based on the promising data from a Phase III study, the combination of relat-limab, an antibody blocking LAG-3 and nivolumab as a fixed-dose combination was recently approved by FDA for the treatment of advanced melanoma as first-line therapy [18].

Major revisions:

Comment 3: My main concern with this manuscript is the lack of novelty, as most of the presented results are not significant, and the correlation between LAG-3/CD8, LAG-3/CTLA-4 and PD-1/LAG-3 has already been described for breast cancer and for many other tumours. I suggest the authors to re-write the paper putting the focus on the novel findings.

Response 3: The reviewer is absolutely correct in that the correlation between LAG-3/CD8, LAG-3/CTLA-4 and PD-1/LAG-3 has already been described in cancer. Nevertheless, our work adds to the existing literature in that we investigated the prognostic significance of LAG-3 mRNA expression in a collective with long follow-up and different molecular subtypes. Furthermore, we report a cohort in which almost half of the patients had not been treated in the adjuvant setting, which allows a clearer estimation of the prognostic significance. For this reason, we prefer to stay with the current and strongly prognosis-driven orientation of our manuscript.

Comment 4: A limitation of the gene expression analysis of your samples is that only 4 single genes were considered for the analysis. It would be interesting if you could study more genes to find more novel correlations.

Response 4: Thanks for this meaningful comment. The reviewer is correct that one of the limitations of our gene expression study is that only 4 genes are studied. In this gene expression analyses we focused on LAG-3 expression. We will evaluate other related genes in further gene expression studies.

Comment 5: Being a retrospectively study, it can be challenging to perform more experimental validations on your data, as the authors discuss. However, it would give more strength to your results to validate them on public omics data, as it would serve as an additional validation. The Gene set enrichment analysis and the validation of the prognostic risk model could be complemented analysing the data with other public databases, which would serve as an additional validation and would give more strength to the conclusions. For example, TIMER2.0 – Cistrome could be used for systematically analysis of immune infiltrates across breast cancer and other cancer types. It would allow the authors to analyse the gene expression for the studied markers markers between tumour and normal, association between gene expression and clinical outcome; correlation between genes, association between immune infiltrates and gene expression; etc.

Response 5: This is a really helpful and valuable comment. In the meantime, we validated the results of our gene expression analyses on a larger, independent cohort and were able to confirm that LAG-3 is a significant prognostic factor in terms of both RFS and OS in the subgroup of triple-negative breast cancers. We have added these validation results to the manuscript to better confirm our initial results:

To validate our gene expression analyses in a larger, independent cohort, we used publicly available gene expression data of LAG-3 with associated tumor characteristics and follow-up data [22] in an unselected cohort of breast cancer patients (n=4929) and in the triple-negative breast cancer subgroup. The prognostic significance of LAG-3 expression was first determined within the overall cohort regarding both RFS and OS. Among the whole cohort LAG-3 expression failed to show prognostic impact both in terms of RFS (p=0.17 Log Rank, n=4929) and OS (p=0.54 Rog Rank; n=1879) (Figure A6a, b). In contrast, within the subgroup of triple-negative breast cancers, LAG-3 was shown to be a significant prognostic factor regarding RFS (p= 0.01, n=335) as well as OS (p=0.037, n=132): higher LAG-3 expression was associated with a better outcome (Figure A7a, b).

Comment 6: Do the authors have access to a QIAGEN Ingenuity Pathway Analysis (IPA) software license? If so, this tool would be very useful to functional analyse, integrate, and further understand data. It has a search capability for information on genes and allows interactive building of networks to represent biological systems. It provides detailed predicted biological activity of your dataset about upstream regulators, causal networks, canonical pathways, multi-levelled regulatory networks, mechanistic networks, downstream effects analysis, tox lists and tox functions, disease, and treatment conditions, among other things. If not, the analysis of the molecular and functional relevance of these genes could be enriched by using some pathway analysis free resources, such as Metascape, Reactome or STRING. Metascape is a tool designed to provide a comprehensive gene list annotation and analysis resource, that can be used to obtain network of enriched terms, top-level Gene Ontology biological processes and interaction networks among other tools. STRING however is a database of known and predicted protein-protein interactions. The interactions include direct (physical) and indirect (functional) associations and could provide interesting hypothetical information for these identified genes at a protein level.

Response 6: thank you for this helpful suggestion. Our manuscript had a focus on the prognostic significance of LAG-3 expression within the overall cohort, as well as within the subgroup of basal-like breast caancers in particular. Our results were validated using an independent, larger cohort (see above). Furthermore, the question of correlations between LAG-3 and the T-cell marker CD8 on the one hand and between LAG-3 and other immune checkpoints (PD-1, CTLA-4) on the other hand was answered. For further assessment of the significance of LAG-3 within the immune microenvironment it may be helpful to investigate downstream effects and mechanistic networks using the proposed tools. This is without question very important, but it would clearly go beyond the scope of our present manuscript.

Comment 7: The authors indicate: “In contrast, in multivariate Cox regression analysis adjusted for age, tumor size, ax-190 illary nodal status, histological grade of differentiation and the proliferation marker Ki-191 67, LAG-3 showed a significant influence on MFS (HR 0.574; 95% CI 0.369–0.894; p = 0.014) 192 (Table 2), a higher LAG-3 expression was associated with a longer MFS.”. Could you show these results as a Figure in addition of the table? It seems that in this sub-section only non-significant results are presented as figures.

Response 7: Thank you for the helpful suggestion. We have presented the data resulting from multivariate Cox regression analyses as forest plots and added them to the manuscript accordingly.

Figure A1: Forest Plot showing LAG-3 as an independent prognostic factor in terms of MFS in multivariate Cox regression analysis adjusted for tumor size, lymph node status, tumor grade, age and the proliferation marker Ki67.

Figure A5: Forest Plot showing CD8 as an independent prognostic factor in terms of MFS in multivariate Cox regression analysis adjusted for tumor size, lymph node status, tumor grade, age and the proliferation marker Ki67.

Point-by-point response to the reviewers’ comments

We appreciate the reviewers’ comments, and we have prepared a revised version of the

manuscript. Please find enclosed our responses to the reviewers’ comments and suggestions:

Reviewer 3

This work describes study the correlation between LAG-3 and metastasis-free survival (MFS) in breast cancer, and the correlation between LAG-3 and CD8 expression.

This study is interesting and has great potential, but also has several flaws:

Minor revisions:

Comment 1: Ethical committee information and patient/participant consent information for patient’s samples should be stated on the Materials and Methods’ section.

Response 1: thank you for the comment. We have extended / supplemented the already existing paragraph in the material and methods part. We added the following paragraph in the section “Material and Methods”:

The study was approved by the Ethics Committee of Rhineland-Palatinate, Germany [no. 837.139.05 (4797)]. Written informed consent was obtained from all patients, and all clinical investigations were conducted ethically in accordance with ethical and legal standards and in consideration of the Declarations of Helsinki.

Comment 2: The authors say: “Further studies will show, if dual blockade of different immune checkpoints should also be recommended in breast cancer.” However, these studies are already being conducted on preclinical and clinical levels. The introduction and discussion section should cite clinical trials that are targeting LAG-3 for breast cancer treatment. That would give strength to the discussion and conclusions. antiPD-1+antiLAG-3 combination have showed encouraging results in many clinical trials. In addition, the first combo (opdualag) has been recently approved by the FDA for clinical use. This should be presented in the introduction.

 Response 2: Thank you for the useful comment. We have supplemented the introduction with the following paragraph:

In addition, based on the promising data from a Phase III study, the combination of relat-limab, an antibody blocking LAG-3 and nivolumab as a fixed-dose combination was recently approved by FDA for the treatment of advanced melanoma as first-line therapy [18].

Major revisions:

Comment 3: My main concern with this manuscript is the lack of novelty, as most of the presented results are not significant, and the correlation between LAG-3/CD8, LAG-3/CTLA-4 and PD-1/LAG-3 has already been described for breast cancer and for many other tumours. I suggest the authors to re-write the paper putting the focus on the novel findings.

Response 3: The reviewer is absolutely correct in that the correlation between LAG-3/CD8, LAG-3/CTLA-4 and PD-1/LAG-3 has already been described in cancer. Nevertheless, our work adds to the existing literature in that we investigated the prognostic significance of LAG-3 mRNA expression in a collective with long follow-up and different molecular subtypes. Furthermore, we report a cohort in which almost half of the patients had not been treated in the adjuvant setting, which allows a clearer estimation of the prognostic significance. For this reason, we prefer to stay with the current and strongly prognosis-driven orientation of our manuscript.

Comment 4: A limitation of the gene expression analysis of your samples is that only 4 single genes were considered for the analysis. It would be interesting if you could study more genes to find more novel correlations.

Response 4: Thanks for this meaningful comment. The reviewer is correct that one of the limitations of our gene expression study is that only 4 genes are studied. In this gene expression analyses we focused on LAG-3 expression. We will evaluate other related genes in further gene expression studies.

Comment 5: Being a retrospectively study, it can be challenging to perform more experimental validations on your data, as the authors discuss. However, it would give more strength to your results to validate them on public omics data, as it would serve as an additional validation. The Gene set enrichment analysis and the validation of the prognostic risk model could be complemented analysing the data with other public databases, which would serve as an additional validation and would give more strength to the conclusions. For example, TIMER2.0 – Cistrome could be used for systematically analysis of immune infiltrates across breast cancer and other cancer types. It would allow the authors to analyse the gene expression for the studied markers markers between tumour and normal, association between gene expression and clinical outcome; correlation between genes, association between immune infiltrates and gene expression; etc.

Response 5: This is a really helpful and valuable comment. In the meantime, we validated the results of our gene expression analyses on a larger, independent cohort and were able to confirm that LAG-3 is a significant prognostic factor in terms of both RFS and OS in the subgroup of triple-negative breast cancers. We have added these validation results to the manuscript to better confirm our initial results:

To validate our gene expression analyses in a larger, independent cohort, we used publicly available gene expression data of LAG-3 with associated tumor characteristics and follow-up data [22] in an unselected cohort of breast cancer patients (n=4929) and in the triple-negative breast cancer subgroup. The prognostic significance of LAG-3 expression was first determined within the overall cohort regarding both RFS and OS. Among the whole cohort LAG-3 expression failed to show prognostic impact both in terms of RFS (p=0.17 Log Rank, n=4929) and OS (p=0.54 Rog Rank; n=1879) (Figure A6a, b). In contrast, within the subgroup of triple-negative breast cancers, LAG-3 was shown to be a significant prognostic factor regarding RFS (p= 0.01, n=335) as well as OS (p=0.037, n=132): higher LAG-3 expression was associated with a better outcome (Figure A7a, b).

Comment 6: Do the authors have access to a QIAGEN Ingenuity Pathway Analysis (IPA) software license? If so, this tool would be very useful to functional analyse, integrate, and further understand data. It has a search capability for information on genes and allows interactive building of networks to represent biological systems. It provides detailed predicted biological activity of your dataset about upstream regulators, causal networks, canonical pathways, multi-levelled regulatory networks, mechanistic networks, downstream effects analysis, tox lists and tox functions, disease, and treatment conditions, among other things. If not, the analysis of the molecular and functional relevance of these genes could be enriched by using some pathway analysis free resources, such as Metascape, Reactome or STRING. Metascape is a tool designed to provide a comprehensive gene list annotation and analysis resource, that can be used to obtain network of enriched terms, top-level Gene Ontology biological processes and interaction networks among other tools. STRING however is a database of known and predicted protein-protein interactions. The interactions include direct (physical) and indirect (functional) associations and could provide interesting hypothetical information for these identified genes at a protein level.

Response 6: thank you for this helpful suggestion. Our manuscript had a focus on the prognostic significance of LAG-3 expression within the overall cohort, as well as within the subgroup of basal-like breast caancers in particular. Our results were validated using an independent, larger cohort (see above). Furthermore, the question of correlations between LAG-3 and the T-cell marker CD8 on the one hand and between LAG-3 and other immune checkpoints (PD-1, CTLA-4) on the other hand was answered. For further assessment of the significance of LAG-3 within the immune microenvironment it may be helpful to investigate downstream effects and mechanistic networks using the proposed tools. This is without question very important, but it would clearly go beyond the scope of our present manuscript.

Comment 7: The authors indicate: “In contrast, in multivariate Cox regression analysis adjusted for age, tumor size, ax-190 illary nodal status, histological grade of differentiation and the proliferation marker Ki-191 67, LAG-3 showed a significant influence on MFS (HR 0.574; 95% CI 0.369–0.894; p = 0.014) 192 (Table 2), a higher LAG-3 expression was associated with a longer MFS.”. Could you show these results as a Figure in addition of the table? It seems that in this sub-section only non-significant results are presented as figures.

Response 7: Thank you for the helpful suggestion. We have presented the data resulting from multivariate Cox regression analyses as forest plots and added them to the manuscript accordingly.

Figure A1: Forest Plot showing LAG-3 as an independent prognostic factor in terms of MFS in multivariate Cox regression analysis adjusted for tumor size, lymph node status, tumor grade, age and the proliferation marker Ki67.

Figure A5: Forest Plot showing CD8 as an independent prognostic factor in terms of MFS in multivariate Cox regression analysis adjusted for tumor size, lymph node status, tumor grade, age and the proliferation marker Ki67.

Point-by-point response to the reviewers’ comments

We appreciate the reviewers’ comments, and we have prepared a revised version of the

manuscript. Please find enclosed our responses to the reviewers’ comments and suggestions:

Reviewer 3

This work describes study the correlation between LAG-3 and metastasis-free survival (MFS) in breast cancer, and the correlation between LAG-3 and CD8 expression.

This study is interesting and has great potential, but also has several flaws:

Minor revisions:

Comment 1: Ethical committee information and patient/participant consent information for patient’s samples should be stated on the Materials and Methods’ section.

Response 1: thank you for the comment. We have extended / supplemented the already existing paragraph in the material and methods part. We added the following paragraph in the section “Material and Methods”:

The study was approved by the Ethics Committee of Rhineland-Palatinate, Germany [no. 837.139.05 (4797)]. Written informed consent was obtained from all patients, and all clinical investigations were conducted ethically in accordance with ethical and legal standards and in consideration of the Declarations of Helsinki.

Comment 2: The authors say: “Further studies will show, if dual blockade of different immune checkpoints should also be recommended in breast cancer.” However, these studies are already being conducted on preclinical and clinical levels. The introduction and discussion section should cite clinical trials that are targeting LAG-3 for breast cancer treatment. That would give strength to the discussion and conclusions. antiPD-1+antiLAG-3 combination have showed encouraging results in many clinical trials. In addition, the first combo (opdualag) has been recently approved by the FDA for clinical use. This should be presented in the introduction.

 Response 2: Thank you for the useful comment. We have supplemented the introduction with the following paragraph:

In addition, based on the promising data from a Phase III study, the combination of relat-limab, an antibody blocking LAG-3 and nivolumab as a fixed-dose combination was recently approved by FDA for the treatment of advanced melanoma as first-line therapy [18].

Major revisions:

Comment 3: My main concern with this manuscript is the lack of novelty, as most of the presented results are not significant, and the correlation between LAG-3/CD8, LAG-3/CTLA-4 and PD-1/LAG-3 has already been described for breast cancer and for many other tumours. I suggest the authors to re-write the paper putting the focus on the novel findings.

Response 3: The reviewer is absolutely correct in that the correlation between LAG-3/CD8, LAG-3/CTLA-4 and PD-1/LAG-3 has already been described in cancer. Nevertheless, our work adds to the existing literature in that we investigated the prognostic significance of LAG-3 mRNA expression in a collective with long follow-up and different molecular subtypes. Furthermore, we report a cohort in which almost half of the patients had not been treated in the adjuvant setting, which allows a clearer estimation of the prognostic significance. For this reason, we prefer to stay with the current and strongly prognosis-driven orientation of our manuscript.

Comment 4: A limitation of the gene expression analysis of your samples is that only 4 single genes were considered for the analysis. It would be interesting if you could study more genes to find more novel correlations.

Response 4: Thanks for this meaningful comment. The reviewer is correct that one of the limitations of our gene expression study is that only 4 genes are studied. In this gene expression analyses we focused on LAG-3 expression. We will evaluate other related genes in further gene expression studies.

Comment 5: Being a retrospectively study, it can be challenging to perform more experimental validations on your data, as the authors discuss. However, it would give more strength to your results to validate them on public omics data, as it would serve as an additional validation. The Gene set enrichment analysis and the validation of the prognostic risk model could be complemented analysing the data with other public databases, which would serve as an additional validation and would give more strength to the conclusions. For example, TIMER2.0 – Cistrome could be used for systematically analysis of immune infiltrates across breast cancer and other cancer types. It would allow the authors to analyse the gene expression for the studied markers markers between tumour and normal, association between gene expression and clinical outcome; correlation between genes, association between immune infiltrates and gene expression; etc.

Response 5: This is a really helpful and valuable comment. In the meantime, we validated the results of our gene expression analyses on a larger, independent cohort and were able to confirm that LAG-3 is a significant prognostic factor in terms of both RFS and OS in the subgroup of triple-negative breast cancers. We have added these validation results to the manuscript to better confirm our initial results:

To validate our gene expression analyses in a larger, independent cohort, we used publicly available gene expression data of LAG-3 with associated tumor characteristics and follow-up data [22] in an unselected cohort of breast cancer patients (n=4929) and in the triple-negative breast cancer subgroup. The prognostic significance of LAG-3 expression was first determined within the overall cohort regarding both RFS and OS. Among the whole cohort LAG-3 expression failed to show prognostic impact both in terms of RFS (p=0.17 Log Rank, n=4929) and OS (p=0.54 Rog Rank; n=1879) (Figure A6a, b). In contrast, within the subgroup of triple-negative breast cancers, LAG-3 was shown to be a significant prognostic factor regarding RFS (p= 0.01, n=335) as well as OS (p=0.037, n=132): higher LAG-3 expression was associated with a better outcome (Figure A7a, b).

Comment 6: Do the authors have access to a QIAGEN Ingenuity Pathway Analysis (IPA) software license? If so, this tool would be very useful to functional analyse, integrate, and further understand data. It has a search capability for information on genes and allows interactive building of networks to represent biological systems. It provides detailed predicted biological activity of your dataset about upstream regulators, causal networks, canonical pathways, multi-levelled regulatory networks, mechanistic networks, downstream effects analysis, tox lists and tox functions, disease, and treatment conditions, among other things. If not, the analysis of the molecular and functional relevance of these genes could be enriched by using some pathway analysis free resources, such as Metascape, Reactome or STRING. Metascape is a tool designed to provide a comprehensive gene list annotation and analysis resource, that can be used to obtain network of enriched terms, top-level Gene Ontology biological processes and interaction networks among other tools. STRING however is a database of known and predicted protein-protein interactions. The interactions include direct (physical) and indirect (functional) associations and could provide interesting hypothetical information for these identified genes at a protein level.

Response 6: thank you for this helpful suggestion. Our manuscript had a focus on the prognostic significance of LAG-3 expression within the overall cohort, as well as within the subgroup of basal-like breast caancers in particular. Our results were validated using an independent, larger cohort (see above). Furthermore, the question of correlations between LAG-3 and the T-cell marker CD8 on the one hand and between LAG-3 and other immune checkpoints (PD-1, CTLA-4) on the other hand was answered. For further assessment of the significance of LAG-3 within the immune microenvironment it may be helpful to investigate downstream effects and mechanistic networks using the proposed tools. This is without question very important, but it would clearly go beyond the scope of our present manuscript.

Comment 7: The authors indicate: “In contrast, in multivariate Cox regression analysis adjusted for age, tumor size, ax-190 illary nodal status, histological grade of differentiation and the proliferation marker Ki-191 67, LAG-3 showed a significant influence on MFS (HR 0.574; 95% CI 0.369–0.894; p = 0.014) 192 (Table 2), a higher LAG-3 expression was associated with a longer MFS.”. Could you show these results as a Figure in addition of the table? It seems that in this sub-section only non-significant results are presented as figures.

Response 7: Thank you for the helpful suggestion. We have presented the data resulting from multivariate Cox regression analyses as forest plots and added them to the manuscript accordingly.

Figure A1: Forest Plot showing LAG-3 as an independent prognostic factor in terms of MFS in multivariate Cox regression analysis adjusted for tumor size, lymph node status, tumor grade, age and the proliferation marker Ki67.

Figure A5: Forest Plot showing CD8 as an independent prognostic factor in terms of MFS in multivariate Cox regression analysis adjusted for tumor size, lymph node status, tumor grade, age and the proliferation marker Ki67.

Round 2

Reviewer 1 Report

No more suggestions.

Author Response

Point-by-point response to the reviewers’ comments

Thank you very much for the useful suggestions and comments, which we have taken into account in the final version of our manuscript. Please find attached a detailed list of the revisions that we have implemented in the current version of the manuscript based on reviewer suggestions.

Point-by-point response to the reviewers’ comments

Thank you very much for the useful suggestions and comments, which we have taken into account in the final version of our manuscript. Please find attached a detailed list of the revisions that we have implemented in the current version of the manuscript based on reviewer suggestions.

Reviewer 2 Report

The authors have addressed some of the comments indicated in the previous review cycle, however, their rebuttal was not reflected on significant changes in the manuscript. For example, the authors should delineate in there Methods section that their cohort were not subject to neoadjuvant treatment, and that protein expression confirmation is not possible due to the reason indicated. The title should strictly indicate that the analysis is limited to gene expression. Other limitations mentioned by the authors in the response letter must be discussed in the Discussion.

Author Response

Point-by-point response to the reviewers’ comments

Thank you very much for the useful suggestions and comments, which we have taken into account in the final version of our manuscript. Please find attached a detailed list of the revisions that we have implemented in the current version of the manuscript based on reviewer suggestions.

Reviewer 2:

Comments:

The authors have addressed some of the comments indicated in the previous review cycle, however, their rebuttal was not reflected on significant changes in the manuscript. For example, the authors should delineate in there Methods section that their cohort were not subject to neoadjuvant treatment, and that protein expression confirmation is not possible due to the reason indicated. The title should strictly indicate that the analysis is limited to gene expression. Other limitations mentioned by the authors in the response letter must be discussed in the Discussion.

Response:

Thank you very much for the meaningful comments.

We have added the following section, highlighted in yellow, to the Material and Methods section to clarify, that the systemic therapies described were applied exclusively as adjuvant chemotherapies and thus had no effect on the mRNA expression of LAG-3 and of further examined immune checkpoints or T-cell associated marker.  

„Patient´s characteristics

The study cohort included 461 patients with early breast cancer who underwent sur-gery at the Department of Gynecology and Obstetrics at the University Medical Center Mainz between 1986 and 2000 and from whom sufficient tumor tissue (fresh frozen) was available for successful Affymetrix microarray analysis. The whole cohort consisted of three subgroups with different systemic treatment:

(i) "N0 cohort": 200 node-negative patients with early breast cancer who received no further adjuvant therapy after surgery and radiation.

(ii) "Tamoxifen cohort": 165 patients treated with tamoxifen as a single adjuvant therapy.

(iii) "Chemotherapy cohort": 96 patients treated with either cyclophosphamide, methotrexate, fluorouracil (CMF; n = 34) or epirubicin, cyclophosphamide (EC; n = 62) in the adjuvant setting. The above mentioned chemotherapy regimens were applied as ad-juvant therapy after completion of surgical therapy and thus had no effect on the analyzed mRNA levels.“

This was also emphasized again in the discussion by the following paragraph (highlighted in yellow):

„A study by Sarradin et al. investigated the effects of neoadjuvant chemotherapy on the composition of the so-called immune microenvironment in triple negative breast cancers [32]. It was shown that LAG-3 expression evaluated by immunohistochemistry decreased significantly after neoadjuvant chemotherapy. LAG-3 expression analyzed before and af-ter completion of chemotherapy had no significant effect on OS [32]. Regarding our retrospective gene expression analysis chemotherapy was applied as adjuvant therapy after completion of surgical therapy and thus had no effect on the analyzed LAG-3 mRNA levels.“

Furthermore, in the discussion, it was added as a potential weakness of the retrospective study that it is limited to gene expression analysis and that the material required for validation at the protein level is not available in sufficient quantity because it is a very old collective.

„Another potential weakness of our study is that our analyses are limited to gene expression data: although it would be interesting and desirable to validate the expression of LAG-3 at the protein level in the same collective, e.g. by immunohistochemistry, possibly in combination with immunofluorescent co-localization studies between LAG-3 and CD8, there is not enough material left to perform such analysis, since this is a very old collective.

To indicate that the retrospective study was limited to gene expression data, the title was modified as follows:

„Prognostic impact of LAG-3 mRNA-expression in early breast cancer“

Reviewer 3:

Comment:  This work remains to be improved in order to be published in Biomedicines.

Response: Thank you for your comment. We have improved a few more points (as listed above) in the final version of the manuscript.

Reviewer 3 Report

This work remains to be improved in order to be published in Biomedicines.

Author Response

(The authors gave the same response as above.)
